# Norepinephrine potentiates and serotonin depresses visual cortical responses by transforming eligibility traces

Su Z. Hong[1], Lukas Mesik[1], Cooper D. Grossman [2], Jeremiah Y. Cohen [2], Boram Lee[3], Daniel Severin[1], Hey-Kyoung Lee [1,2], Johannes W. Hell [3] & Alfredo Kirkwood [1,2 ✉]

Reinforcement allows organisms to learn which stimuli predict subsequent biological relevance. Hebbian mechanisms of synaptic plasticity are insufficient to account for reinforced learning because neuromodulators signaling biological relevance are delayed with respect to the neural activity associated with the stimulus. A theoretical solution is the concept of eligibility traces (eTraces), silent synaptic processes elicited by activity which upon arrival of a neuromodulator are converted into a lasting change in synaptic strength. Previously we demonstrated in visual cortical slices the Hebbian induction of eTraces and their conversion into LTP and LTD by the retroactive action of norepinephrine and serotonin Here we show in vivo in mouse V1 that the induction of eTraces and their conversion to LTP/D by norepinephrine and serotonin respectively potentiates and depresses visual responses. We also show that the integrity of this process is crucial for ocular dominance plasticity, a canonical model of experience-dependent plasticity.

[1] Mind/Brain Institute, Johns Hopkins University, Baltimore, MD 21218, USA. [2] Department of Neuroscience, Johns Hopkins University, Baltimore, MD 21205, USA. [3] Department of Pharmacology, University of California at Davis, Davis, CA 95616, USA. ✉email: Kirkwood@jhu.edu

A fundamental role of our brain is to learn stimuli predicting biologically relevant values, such as novelty, saliency, and reward. Neuromodulator signals have been thought to encode these value signals and teach the system which input should be reinforced[1,2]. However, the evidence of biological value is often only available with temporal delay, which raises a question known as the credit assignment problem[3,4]: How does the delayed value signal gate the plasticity in synapses that were transiently activated by the predictive stimulus? As a theoretical solution to bridge the temporal gap between stimulus and value signal, synaptic eligibility traces (eTraces) have been hypothesized. The eTrace is a transient and silent process, which is generated by the pre- and postsynaptic activities of a neuron. The eTrace can last until the arrival of a neuromodulatory signal, forming a bridge between neuronal activities and behaviorally relevant neuromodulatory signals. This process allows selective plasticity of the synapses relevant to the biological values, while at the same time bridging the temporal delay that commonly exists between neuronal activity and feedback regarding its behavioral relevance[5–9].

Synaptic eTraces were experimentally demonstrated first in the mushroom body of insects[10], and later in several in vitro preparations of the mammalian brain, including striatum[11,12], hippocampus[13,14], and neocortex[15]. Particularly, we have shown the existence of two distinct eligibility traces (eTraces), the LTP and LTD traces, in primary visual cortex (V1) slices[15]. Specifically, the pre- and postsynaptic activities generate the LTP trace at the synapse, which is converted into functional synaptic LTP by norepinephrine via involvement of β2-adrenergic receptors (β2AR) in layer 2/3 V1 pyramidal neurons. On the other hand, the post and then presynaptic activities generate the LTD trace, which is converted into synaptic LTD by serotonin via 5HT2c receptors (5HT2cR)[15]. Notably, in contrast to several demonstrations of the eTraces in vitro, in slice preparations, the evidence of neuronal plasticity mediated by the transformation of the eTraces in vivo is limited only to LTP studies and only in the striatum[16,17]. Here we show that the transformation of both LTP and LTD traces in V1 in vivo, respectively, potentiates or depress visual cortical responses and that the integrity of this reinforced-like mechanism is crucial for the canonical model of cortical modification, ocular dominance plasticity.

## Results

Previously we demonstrated in vitro the Hebbian induction of synaptic eTraces and its conversion into LTP and LTD by adrenergic and serotonergic receptors, respectively. Here we investigated the functional relevance of eTraces in vivo in the mouse visual cortex. To that end we tested 1) whether the release of neuromodulators timed in a reinforced-like paradigm modifies visual responses, ocular dominance and receptive field selectivity 2) whether Hebbian associative paradigms can induce eTraces, and 3) whether blocking the conversion of eTraces into LTP and LTD prevents ocular dominance plasticity.

**Potentiation and depression of visual cortical responses by the retroactive action of norepinephrine and serotonin.** We first tested whether the timely release of norepinephrine (NE) or serotonin (5HT) after visual stimulation potentiates or depresses visual responses in V1 in a manner consistent with the transformation of eTraces. To that end, we quantified visual cortical responses before and after conditioning with the optogenetically-induced release of neuromodulators in mice expressing ChR2 in either noradrenergic (NE-ChR2) or serotonergic neurons (5HT-ChR2) (see "Methods"). Visual responses were elicited by drifting horizontal and vertical bars presented to one eye and recorded in the contralateral V1 using optical imaging of the intrinsic signal

(ISI) (Fig. 1a). Conditioning consisted of the alternating presentation of full-field horizontal and vertical drifting gratings, where the horizontal presentations, but not the vertical ones, were immediately followed by direct LED illumination of the exposed cortex (5 s train of 470 nm pulses of 10 ms at 20 Hz; Fig. 1b).

In NE-ChR2 mice, this conditioning resulted in substantial potentiation of the responses to the horizontal orientation, without changes in the responses to the unpaired vertical orientation (Fig. 1c, d). As a result, the ratio of horizontal versus vertical response amplitude (HV ratio), typically biased toward the horizontal orientation in naïve mice[18,19], was further increased after conditioning. In contrast, in the 5HT-ChR2 mice, the conditioning resulted in the selective depression of the horizontal responses without changes in the vertical responses and, consequently, a reduction in the HV ratio (Fig. 1e, f). Thus, the response to the same stimulation can be potentiated or depressed, depending on the neuromodulator released. As a control for the input specificity of the induced changes, we verified that the vertical responses are also potentiated and depressed by reinforcement-like conditioning (Supplementary Fig. 1). In addition, we confirmed that conditioning the responses to the contralateral eye potentiates or depresses these responses, depending on the neuromodulator released, but it did not affect the responses evoked by the ipsilateral, non-conditioned, eye (Supplementary Fig. 2). Altogether, the results are consistent with a scenario in which visual stimulation induces eTraces that can be converted to LTP and LTD by NE and 5HT, respectively.

To test the involvement of eTraces more directly, we exploited our previous observation, made in vitro, that the conversion of eTraces into LTP and LTD by NE and 5HT requires the anchoring of β-adrenergic (βAR) and 5HT$_{2c}$ serotonergic receptors to the postsynaptic protein PSD95[15]. Thus, peptides that mimic the C-terminal of these receptors disrupt their synaptic anchoring and prevent the conversion of the eTraces[15]. We tested, therefore, whether intraventricular infusion of cell-permeable versions of these peptides prior to the experiments prevents the modification of the visual responses by the reinforcement-like conditioning described above (Fig. 2a–d). In the NE-ChR2 mice injected with the peptide DSPL to disrupt the anchoring of βAR, the conditioning did not affect the HV ratio, whereas in mice injected with the control peptide DAPA[15] the conditioning did increase the HV ratio (Fig. 2e). In a similar fashion, in 5HT-ChR2 mice injected with the peptide 2C-ct (TAT version of VNPSSVVSERISSV[15]) to disrupt 5HT$_{2c}$R, the conditioning failed to affect the HV ratio, yet the conditioning effectively reduced the HV ratio in mice injected with the control CSSA peptide (Fig. 2f). We considered the possibility that prolonged exposure to the peptides, which do not affect the expression of LTP and LTD[15], might affect the regulation of intrinsic cell excitability by 5HT$_{2c}$ and β-adrenergic receptors[20] and compromise the interpretation of the results. Arguing against that possibility, we found that in slices the application of the peptides did not affect the increase in layer 2/3 cell firing induced by 5HT$_{2c}$ and β-adrenergic agonists (Supplementary Fig. 2). This result also suggests that the conversion of eTraces, but not the regulation of pyramidal cell excitability, is dependent on the anchoring of these receptors to the PSD. Although we cannot exclude off-target effects of the peptides, the results are consistent with the notion that the visual stimulation-induced eTraces were subsequently converted into LTP and LTD by the retroactive action of NE and 5HT, respectively.

**Potentiation and depression of cortical visual responses by Hebbian induction and optogenetic conversion of eligibility traces.** We previously demonstrated in vitro the induction of

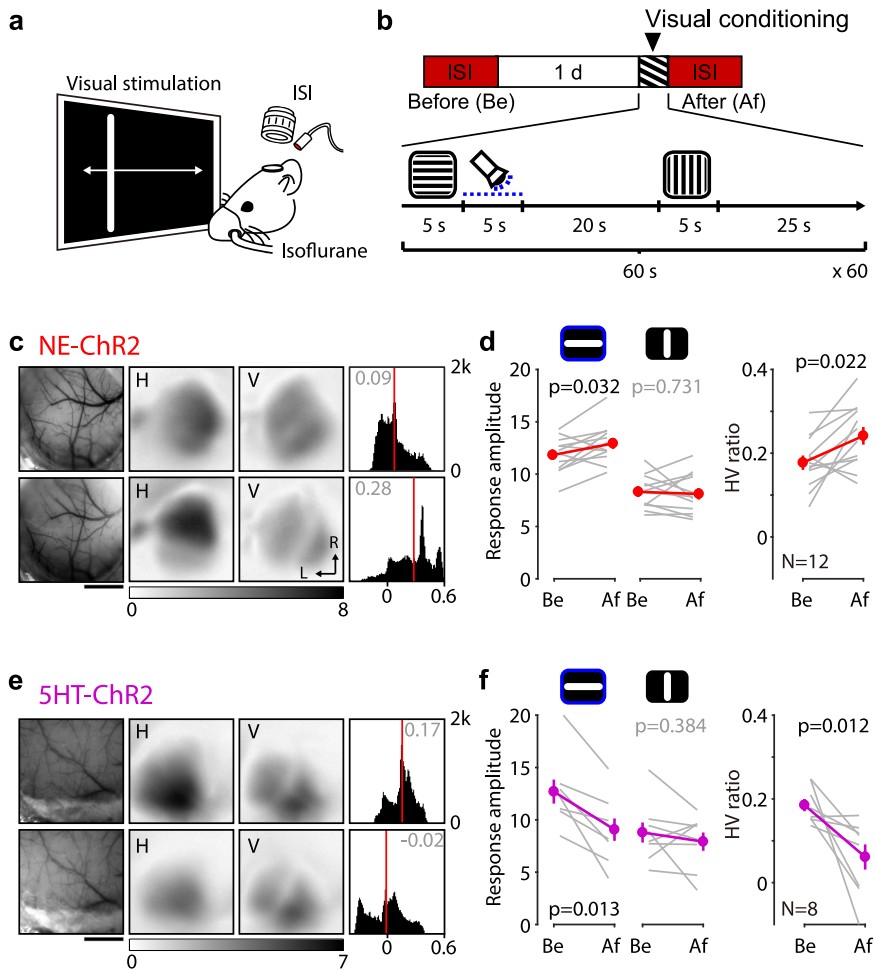

**Fig. 1 Potentiation and depression of visual cortical responses by the retroactive action of norepinephrine and serotonin. a** Schematic of the optical imaging of the intrinsic signal (ISI) of the visual cortical response from the V1. **b** Experiment timeline (top) and the visual conditioning protocol (bottom). Blue dotted line indicates photoactivation of ChR2. **c, d** Optogenetic transformation of LTP trace induces potentiation of the associated visual cortical response of the NE-ChR2 mice. **c** Representative change of the visual cortical response by the visual conditioning. Left: vasculature pattern of the imaged region used for alignment. Scale bar, 1 mm. Middle: magnitude map of the visual cortical response evoked by horizontal [H] or vertical [V] drifting bar. Gray scale (bottom): response magnitude as the fractional change in reflection x10$^4$. Arrows: L lateral, R rostral. Right: histogram of HV ratio illustrated in the number of pixels (x-axis: HV ratio, y-axis: number of pixels). **d** Summary of changes in response amplitude evoked by the horizontal (left) or vertical (middle) drifting bar as well as the change of HV ratio (right) before (Be) and after (Af) the conditioning. Thin line: individual experiments; thick line and symbols: average ± s.e.m. **e, f** Optogenetic transformation of LTD trace induces depression of the associated visual cortical response of the 5HT-ChR2. Same format with (c and d). Source data are provided as a Source Data file.

eTraces with Hebbian paradigms that associate pre- and postsynaptic activation. To examine that possibility in vivo, we tested whether pairing subthreshold visually evoked postsynaptic potentials (VEPSPs) with postsynaptic firing can result in LTP or LTD if followed by optogenetic delivery of endogenous monoamines in the NE-ChR2 and 5HT-ChR2 mice. We performed in vivo whole-cell recordings from excitatory neurons in the superficial layers of V1 (Fig. 3 and Supplementary Fig. 3, see "Methods") and subthreshold VEPSPs were evoked in the same cell by two alternating rectangular flashing lights (500 ms) selected from 15 non-overlapping subregions of a screen (Fig. 3a). During associative Hebbian conditioning each VEPSP was paired with a long burst of postsynaptic spikes (induced by current injection; 400 ms, 726.6 ± 170.1 pA, 10.9 ± 3.3 spikes, mean ± s.d.; Fig. 3d) that preceded and overlapped with the visual stimulation. The purpose of this temporal arrangement was to ensure post-pre and pre-postsynaptic associations that we had shown in vitro to induce eTraces for LTP and LTD[15]. During the conditioning, optogenetic release of neuromodulators (1 s train of 470 nm LED

pulses, 10 ms at 20 Hz) was immediately coupled to one of the VEPSPs (C-VEPSP), the other VEPSP uncoupled to neuromodulator release (U-VEPSP) served as an internal control.

To test the transformation of LTP traces, we recorded in NE-ChR2 mice (TH-ChR2, see "Methods") and the optogenetic release of NE was evoked by direct illumination of the cranial window (Supplementary Fig. 3a, b, see "Methods"). In accord with our previous in vitro findings[15] after the conditioning, the C-VEPSPs were potentiated ($p = 0.006$; Fig. 3f, g). The U-VEPSPs, on the other hand, were slightly, but significantly depressed ($p = 0.022$) after the visual conditioning (Fig. 3f, g). These changes were activity-dependent because the optogenetic activation alone did not affect the VEPSPs (104.4 = 7.8 %, $p = 0.635$, $n = 13$) (Supplementary Fig. 3f, g). In addition, the conditioning did not affect the passive membrane properties of the cells (Supplementary Fig. 3d)

We examined the transformation of LTD traces by illuminating, via optic fiber, the dorsal raphe nucleus (DRN) of 5HT-ChR2 mice (Sert-Cre, Supplementary Fig. 3c, see "Methods").

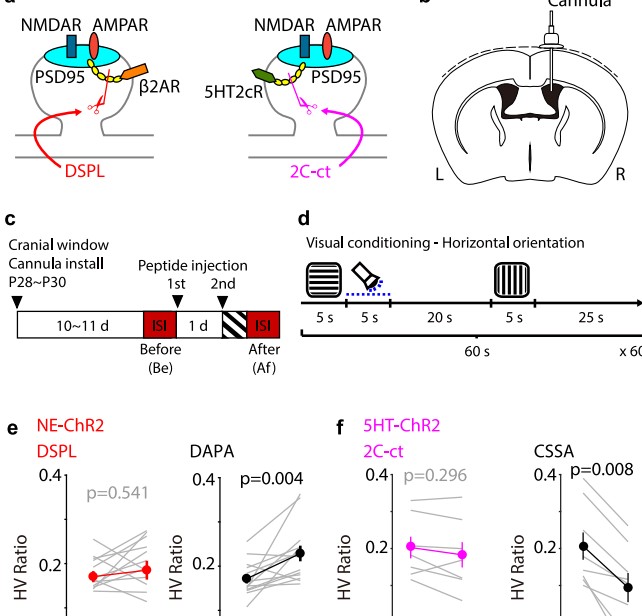

**Fig. 2 Peptides targeting the conversion of eTraces prevent reinforcement-like conditioning of visual cortical responses. a** Diagrams illustrating that DSPL (left) or 2C-ct (right) disrupts the direct interaction of the β2AR or 5HT2cR with PSD-95, respectively. **b** A diagram illustrating the implantation of the cannula to inoculate the disrupting peptides to the lateral ventricle. **c** Experiment timeline. The peptide was injected 1 day (1st) and 30 min (2nd) earlier than the visual conditioning. **d** For the visual conditioning, horizontal or vertical drifting gratings were alternately shown at 30 s intervals. Photoactivation to induce the release of neuromodulators was retroactively coupled to the horizontal drifting gratings. **e** Summary of the HV ratio change by the visual conditioning in the presence of the disrupting peptide (DSPL) or the control peptide (DAPA) of the NE-ChR2 mice. **f** Summary of the HV ratio change by the visual conditioning in the presence (2C-ct) or the control peptide (CSSA) of the 5HT2cR disrupting peptide of the 5HT-ChR2 mice. Thin line: individual animals; thick line and symbols: average ± s.e.m. Source data are provided as a Source Data file.

Consistent with our previous slice results[15], the conditioning depressed the C-VEPSPs (NC: 25.8 = 7.9%, $p = 0.016$, $n = 12$) without affecting the U-VEPSPs (NU: 0.3 = 4.4%, $p = 0.85$, $n = 12$) (Fig. 3h, i). Together, the results in the 5HT-ChR2 and NE-ChR2 mice demonstrate in vivo that the conjunction of synaptic activation and postsynaptic firing is sufficient to induce eTraces that subsequently can be converted into potentiation or depression by the retroactive actions of NE and 5HT.

**Optogenetic transformation of eligibility traces changes the orientation responses of individual V1 neurons.** A common feature of V1 neurons is their orientation selectivity, which is highly modifiable by sensory experience. We asked, therefore, whether reinforcement with optogenetic delivery of NE and 5HT can affect the orientation responses of individual V1 neurons. On day 1 we recorded responses of the cells to drifting gratings of 6 orientations (in two directions) with two-photon calcium imaging in mice virally infected to express GCamp6f (see Fig. 4a, b and Supplementary Fig. 4). On day 2 the conditioning took place and the orientation response was determined again. During conditioning, one orientation (which varied from mouse to mouse. See "Methods") was paired with the optogenetic release of neuromodulators in a similar fashion as it was done with the Hebbian

studies (Fig. 4b). The orthogonal orientation, which was not paired, was also delivered in an alternated fashion and served as a control.

We examined the potentiation of orientation responses by a reinforcement-like paradigm in 57 responsive cells recorded from 11 head-fixed awake NE-ChR2 mice (Fig. 4c–i). An example cell is shown in Fig. 4c–e. Before conditioning the cell responded best to a 60° orientation, after conditioning the cell became most responsive to the 90° used in the pairing (Fig. 4c, d). This particular cell had a clear orientation preference, which shifted towards the reinforced orientation after the conditioning; a change that was quantified as the angular difference between the preferred and the conditioned orientation (Fig. 4e). In the total population of examined cells, on average we observed an increased response that was specific to the conditioned orientation (Fig. 4f, g). In the subset of 31 clearly orientation-selective cells, the preferred orientation shifted towards the conditioned one (Fig. 4h), without losing overall selectivity (Fig. 4i), characterized as 1-CirVar (see "Methods"). Those changes were not observed in control experiments in which the LED illumination was omitted during the visual conditioning (z-score before: $0.09 \pm 0.22$; after: $0.09 \pm 0.21$; 3 mice, 22 cells; $p = 0.997$).

In 5HT-ChR2 mice we examined the depression of individual cell responses by the reinforcement-like paradigm. Initial experiments, performed as in Fig. 4c–e in head-fixed awake mice, resulted in no changes in the response amplitude to the conditioned orientation (z-core before: $0.99 \pm 0.15$; after: $0.93 \pm 0.15$; 3 mice, 32 cells; $p = 0.515$) even after a fluoxetine injection to reduce 5HT uptake (3 mice, 21 cells; $p = 0.252$). Therefore, we switched to the anesthetized preparation because it worked for the experiments previously described in Figs. 1, 2. Under these conditions, the pairing with the optogenetic stimulation of the dorsal raphe nucleus did result in the selective depression of the paired orientation. An individual example is shown in Fig. 4j–l; the summary averages of 62 cells in 6 mice are shown in Fig. 4m–p. It is presently unclear .why the reinforcement-like paradigm with the current timing parameters was effective only in the anesthetized, and not in awake 5HT-ChR2 mice. Possibly, this might relate to elevated levels of 5HT or an elevated threshold for depressing mechanisms in the head-fixed awake mice. Nevertheless, collectively the results indicate that, in vivo, response preferences of V1 cells can be shifted away from or towards a targeted orientation by the retroactive action of 5HT and NE, respectively.

**Preventing the transformation of eTraces blocks ocular dominance plasticity.** In Figs. 1–4 we showed that optogenetically released NE and 5HT after visual stimulation can modify cortical responses in a manner consistent with the induction and conversion of eTraces. Because these results were obtained in anesthetized or head-fixed mice, it was of interest to examine the role of eTraces in cortical plasticity occurring in non-constrained, freely behaving mice. We focused on ocular dominance plasticity (ODP) induced by monocular deprivation (MD), a canonical model of sensory-induced cortical modification that is well established in mice. In juvenile mice, MD results in an initial rapid depression of cortical responsiveness to the deprived eye; whereas at later ages, in young adults, MD manifests as a delayed potentiation of the responses to the non-deprived eye[21–23]. These changes have been attributed to LTD and LTP mechanisms[24] (but see Turrigiano and Nelson[25]). We asked, therefore, how the 2C-ct and the DSPL peptides, respectively, targeting the synaptic anchoring of 5HT2c and β-adrenergic receptors, affect ODP. The changes in cortical response to the two eyes were monitored with intrinsic signal imaging in the binocular zone of V1 contralateral to the deprived eye[26].

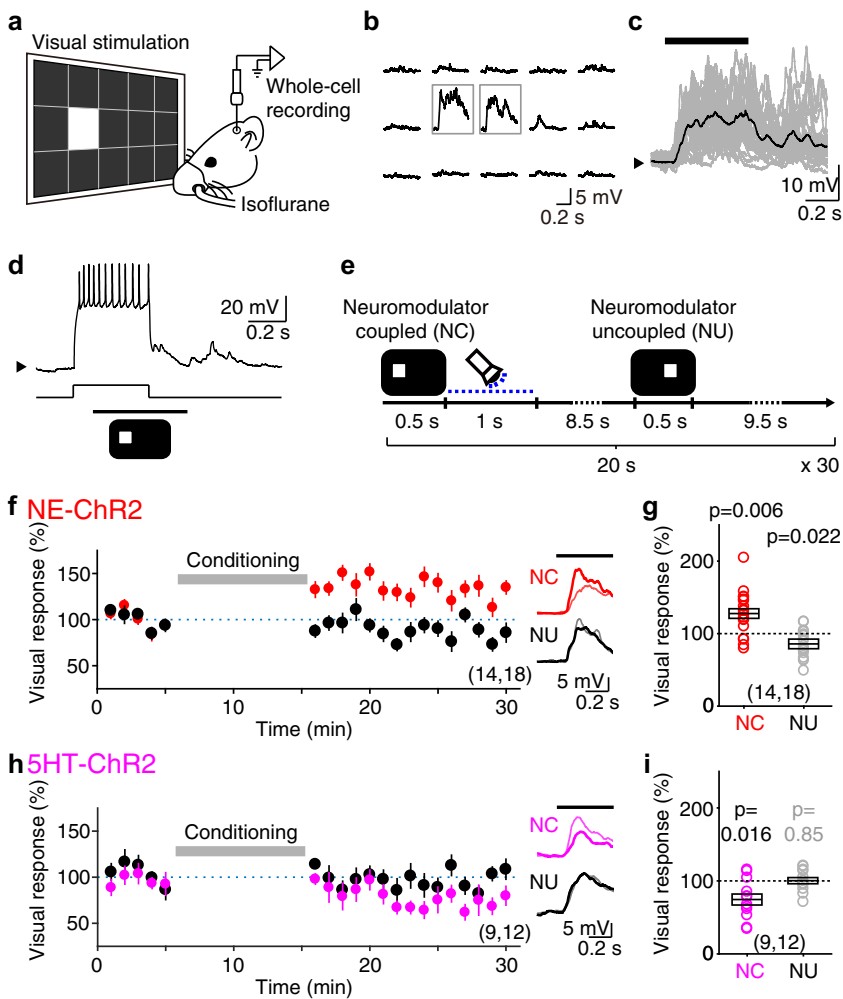

**Fig. 3 Potentiation and depression of VEPSPs by Hebbian induction and optogenetic conversion of eligibility traces. a** Schematic of in vivo whole-cell patch-clamp recording of the superficial V1 neurons. **b** Example of the VEPSPs elicited by visual stimuli at each subregion of a screen. Gray boxes indicate the two panels chosen for visual stimulation. **c** Individual (gray traces) and averaged (black trace) VEPSPs elicited by the visual stimulus presentation (black bar on top) of a representative neuron. Black arrow on the left indicates Vm (−70 mV). **d** Pairing of VEPSP with a burst of postsynaptic spikes. The current injected via recording pipette and the timing of visual stimulus are shown at the bottom. Black arrow on the left indicates Vm (−70 mV). **e** Visual conditioning protocol. **f** Normalized change of the NC (red) or NU (black) VEPSP amplitude of the NE-ChR2 mice by the visual conditioning. Inset traces show the VEPSPs of a representative neuron averaged initial (thin line) or last (thick line) 5 min of the recording. **g** Summary of VEPSP changes by the visual conditioning. Box plot: average ± s.e.m. Sample number indicates the number of animals and the number of recorded neurons. **h**, **i** Same as panel (**f**, **g**) but for the 5HT-ChR2 mice. Source data are provided as a Source Data file.

First, we tested whether the MD-induced depression of the deprived eye in juvenile mice (p28-29) requires the transformation of LTD trace and continuously infused the 2C-ct peptide, or its CSSA control, in the lateral ventricle starting 1 day prior to the eye-suture (Fig. 5a–c). As expected from the previous studies[21,23], in control mice infused with the control CSSA peptide, 3 days of MD selectively reduced the responses to the deprived-eye (contralateral) without affecting the responses to the non-deprived eye (ipsilateral) (Fig. 5d, f, g). In contrast, in mice infused with the 2C-ct peptide, the brief MD-induced negligible changes in the responses to either eye (Fig. 5e–g). A two-way ANOVA (Peptide x MD) and *post hoc* comparisons confirmed the significance of the interaction between the peptides and MD in the contralateral deprived-eye response ($F(1, 18) = 15.2$, $p = 0.001$) as well as in the ocular dominance index (ODI: see "Methods") ($F(1, 18) = 22.63$, $p < 0.001$). Thus, the normal MD-induced shift in response balance towards the non-deprived eye, quantified as a reduction in ocular dominance index (ODI: see

"Methods"), failed to occur in CSSA-infused mice (Fig. 5h). These results support the idea that conversion of eTraces for LTD by 5HT2c receptors is necessary for juvenile ODP.

Next, we tested whether the transformation of the LTP trace is required for potentiating the non-deprived eye during prolonged MD (7 days) in young adult mice (p90-96) and evaluated the effects of infusing the DSPL peptide and the control DAPA peptide (Fig. 6). In these studies, infused and non-infused adult mice (p90–96) were subjected to MD for 7 days with the ocular dominance analyzed at the end of MD (Fig. 6a, b). As expected from previous studies, compared to the age-matched normal reared mice, non-infused mice with 7d MD showed on average a potentiated response to the non-deprived eye. Consequently, their ODI was reduced, reflecting the shift toward the open eye (Fig. 6c, d). In contrast, mice infused with the DSPL peptide showed negligible potentiation of the non-deprived eye as well as minimally reduced ODI (Fig. 6g–i). On the other hand, mice with the control peptide DAPA did show the open-eye potentiation and reduced ODI comparable to the one

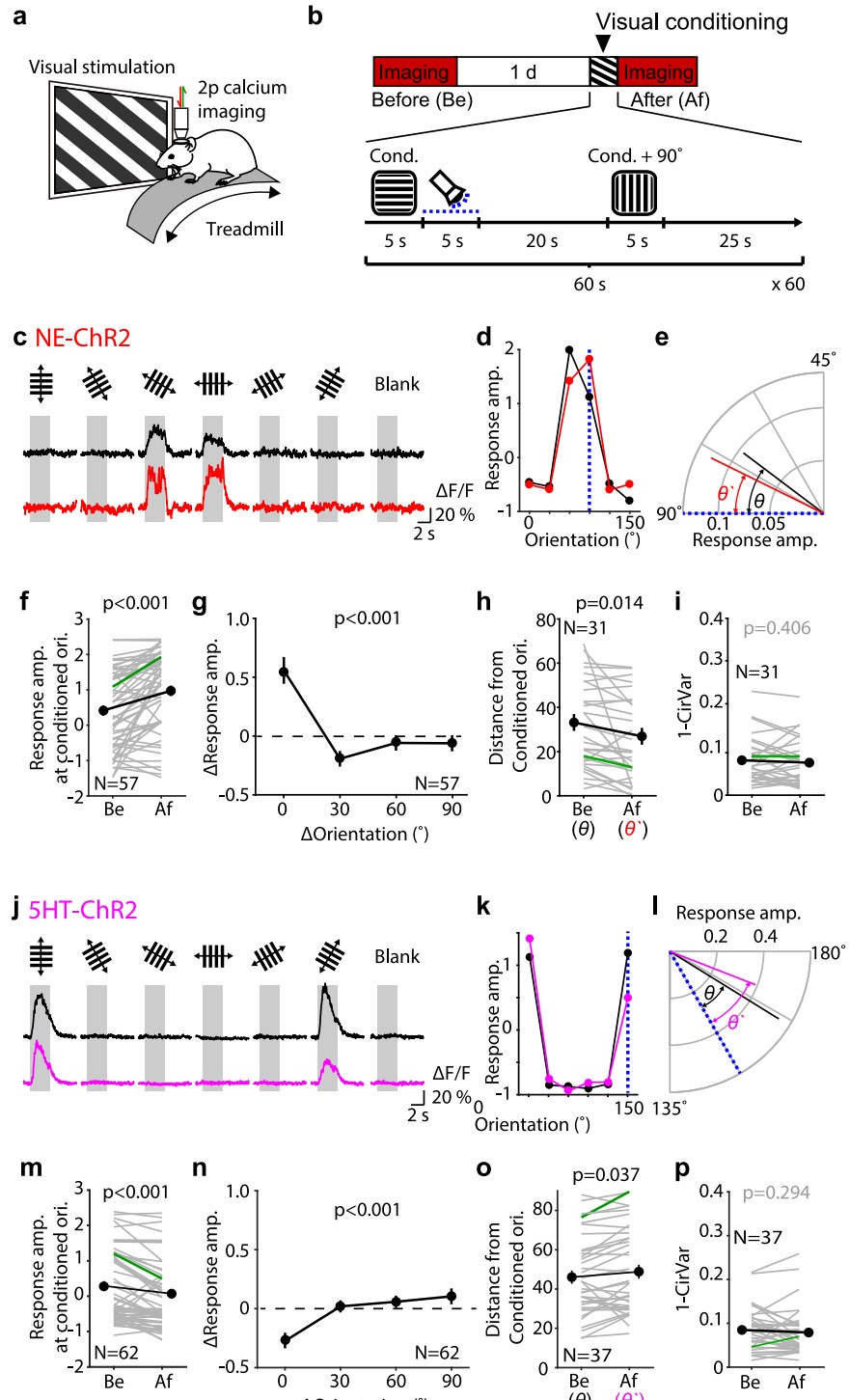

registered in non-infused 7dMD mice (Fig. 6g–i). These results imply a critical role of the transformation of LTP trace in the potentiation of the open eye during the ODP.

Finally, to verify the specificity of the disrupting peptides in the interruption of the LTP or LTD trace transformation, we cross-examined the actions of the peptides and asked whether the 2C-ct affects the potentiation of the open eye in adult mice and whether DSPL impairs the depression of closed eye in juvenile mice. In adult mice infused with 2C-ct, the non-deprived eye potentiation induced by 7d MD was normal; and in juvenile mice infused with DSPL, the depression of the deprived eye after 3d MD was also normal (Supplementary Fig. 5). This confirmation of the specific

action of the peptides further validates the idea that the induction of eTraces and their conversion into LTP and LTD is a necessary process in ODP.

## Discussion

Our results provide the first direct evidence that eTraces for LTP and LTD play a role in experience-dependent cortical plasticity. We demonstrated the Hebbian induction of these eTraces (Fig. 3) and also showed that their transformation by the retroactive action of NE and 5HT is sufficient to potentiate or depress visual responses in V1 (Figs. 1, 2). Moreover, optogenetic recruitment of this reinforcement-like mechanism mimics the changes in ocular

**Fig. 4 Optogenetic transformation of eligibility traces changes orientation responses of individual V1 neurons. a** Schematic of the two-photon calcium imaging of head-fixed mice. **b** Experiment timeline (top) and the visual conditioning protocol (bottom). Blue dotted line indicates photoactivation of ChR2. **c–e** Analysis of a representative neuron from a NE-ChR2 mouse. **c** Fluorescence signal elicited by various orientations of the drifting gratings before (black) and after (red) the visual conditioning. Each trace indicates the average response across 16 trials consisting of both directions. Gray area indicates the visual stimulus. **d** Orientation tuning curve before (black) and after (red) the visual conditioning. Blue dotted line indicates the conditioned orientation. **e** Vectors indicating the preferred orientation before (black) and after (red) the visual conditioning. Blue dotted line indicates the conditioned orientation. $\theta$ and $\theta`$ indicate the angular differences between the preferred orientations and the conditioned orientation. **f–i** Summary of the changes in NE-ChR2 mice. **f** Change of the response amplitude at the conditioned orientation. **g** Response amplitude change according to the difference from the conditioned orientation. Data at two orientations (clockwise and counterclockwise) were pulled for comparison. (Kruskal–Wallis test, KW stat = 42.58, $p < 0.001$, and post hoc Dunn's multiple comparison test (vs. 0° $\Delta$Orientation, 30°: $p < 0.001$, 60°: $p = 0.005$, 90°: $p = 0.001$). **h** Angular difference between the preferred orientations and the conditioned orientation measured before and after the visual conditioning diminished. This analysis includes only oriented cells with initial preferred direction significantly different from the conditioned one. **i** 1-CirVar as a measure of orientation selectivity of the cells measured before and after visual conditioning (see "Methods"). Thin gray lines in (**h**, **i**): individual neurons; thick line and symbols: average ± s.e.m. Green lines indicate the example neuron in (**c–e**). **j–p** Summary of the changes in 5HT-ChR2 mice. Same format with (**f–i**). (Kruskal–Wallis test, KW stat = 30.59, $p < 0.001$, and post hoc Dunn's multiple comparison test (vs. 0° $\Delta$Orientation, 30°: $p < 0.001$, 60°: $p < 0.001$, 90°: $p < 0.001$). Source data are provided as a Source Data file.

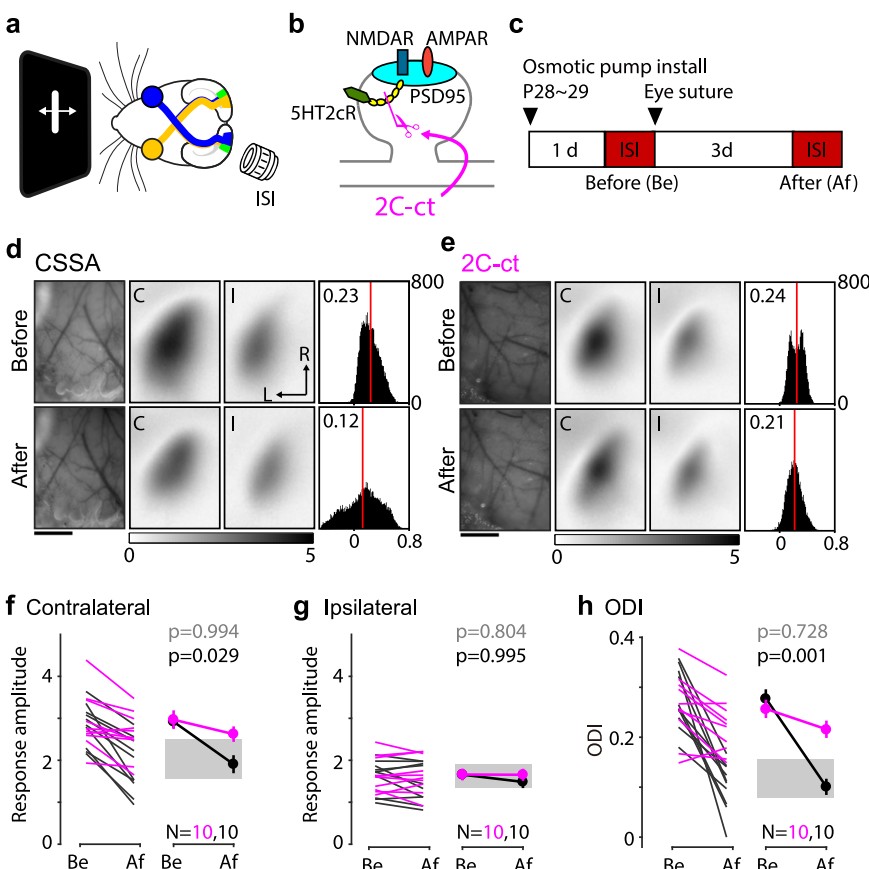

**Fig. 5 Blockage of the conversion of the LTD trace impairs the ocular dominance plasticity in juvenile mice. a** Schematic of the experiment to record the visual cortical response in the binocular region (green region in left hemisphere) of the V1. **b** A diagram illustrating that 2C-ct disrupts the direct interaction of the 5HT2c receptor with PSD-95. **c** Experiment timeline. ISI was performed before (Be) and after (Af) the 3d MD. **d**, **e** Representative change of the visual cortical response by the 3d MD in the presence of the control peptide, CSSA (**d**), or 2C-ct (**e**). Left: vasculature pattern of the imaged region used for alignment. Scale bar, 1 mm. Middle: magnitude map of the visual cortical response evoked by contralateral [C] or ipsilateral [I] eye from the recorded hemisphere. Gray scale (bottom): response amplitude as the fractional change in reflection x10[4]. Arrows: L, lateral, R, rostral. Right: histogram of the ODI illustrated in the number of pixels (x-axis: ODI, y-axis: number of pixels). Red line indicates the average. **f–h** Summary of the changes in response amplitude evoked by the contralateral (**f**) and ipsilateral (**g**) eye as well as the change of ODI (**h**) before (Be) and after (Af) the conditioning. Left: individual experiments in the presence of CSSA (black) or 2C-ct (purple); Right: average ± s.e.m. of the left plot. *p*-values: Two-way ANOVA and post hoc Sidak's multiple comparisons test between the 2C-ct group and the CSSA group at before (gray) and after(black) the 3d MD. Gray region indicates a 95% confidential interval of 3d MD mice without peptide infusion. Source data are provided as a Source Data file.

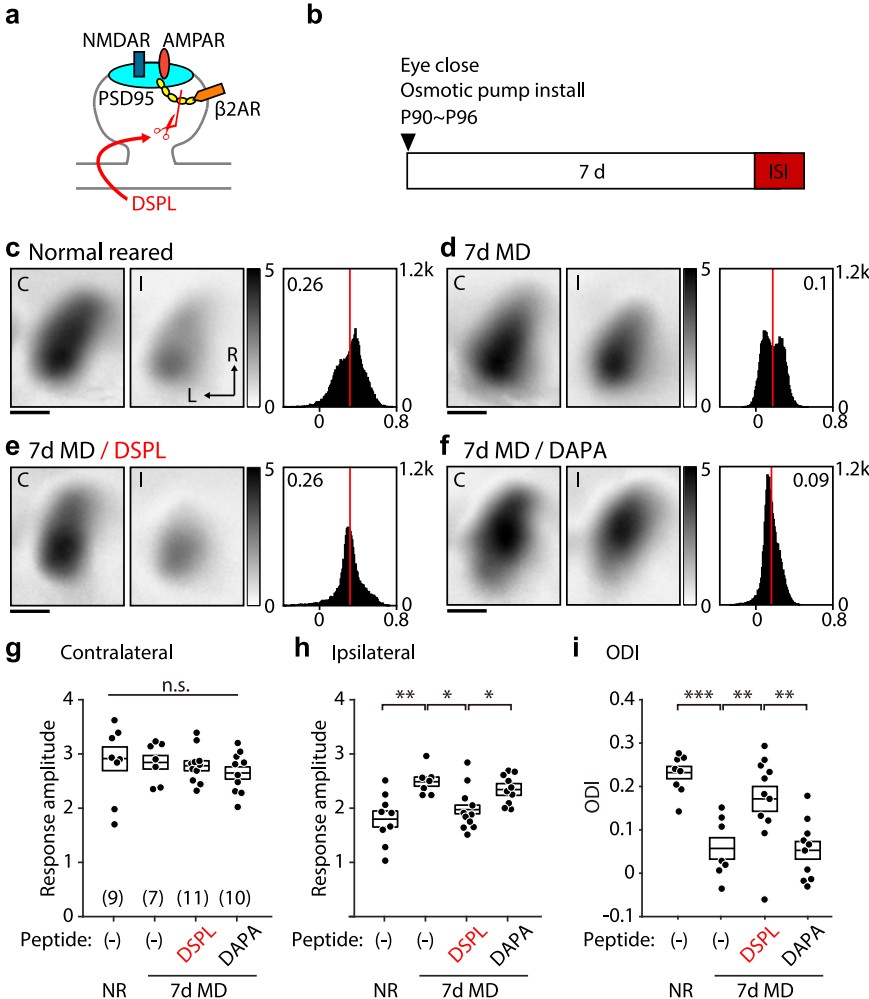

**Fig. 6 Blockage of the conversion of the LTP trace impairs the ocular dominance plasticity in young adult mice. a** A diagram illustrating that DSPL disrupts the direct interaction of the β2-adrenergic receptor with PSD-95. **b** Experiment timeline. For the MD, the contralateral eye to the recorded hemisphere was closed for 7 days before the ISI. Peptide infusion was started when the eye closed. **c–f** Representative visual cortical response of the mouse normal reared (**c**), after 7 days of MD (**d**), after 7 days of MD with the infusion of DSPL (**e**), and after 7 days of MD with the infusion of the control peptide, DAPA (**f**). Left and middle: each magnitude map shows the visual cortical response from the contralateral (C) or the ipsilateral (I) eye from the recorded hemisphere. Gray scale: response amplitude as the fractional change in reflection ×10$^4$. Arrows: L, lateral, R, rostral. Right: histogram of the ODI illustrated in the number of pixels (x-axis: ODI, y-axis: number of pixels). **g–i** Summary of the response amplitude evoked by the contralateral (**g**) and ipsilateral (**h**) eye as well as the ODI (**i**) of each group. Box plot: average ± s.e.m. One-way ANOVA and post hoc Holm–Sidak's multiple comparison test, **g** $F_{(3,33)} = 0.578$, $p = 0.633$; **h** $F_{(3,32)} = 6.267$, $p = 0.001$; **i** $F_{(3,33)} = 12.31$, $p < 0.001$; *$p < 0.05$, **$p < 0.01$, ***$p < 0.001$. Source data are provided as a Source Data file.

dominance induced by monocular deprivation (Supplementary Fig. 2), whereas disrupting it prevents the effects of monocular deprivation (Figs. 5, 6). These findings line up with the emerging notion of a "three-factor rule" for synaptic plasticity stating that besides STDP-like associations of pre- and postsynaptic activity, the expression of synaptic plasticity is also dependent on neuromodulators signaling behavioral relevance[2,9].

The essential role of neuromodulation in cortical plasticity is well established since the initial finding that ablation of noradrenergic function prevents ocular dominance plasticity[27]. Subsequent studies extended these observations to other neuromodulatory systems and other sensory cortices[28–33]. Initially, neuromodulators were thought of as permissive/enabling factors that promote the induction of Hebbian synaptic plasticity via the enhanced cellular and network excitability associated with arousal and the awake state. It was later found that besides facilitating it, specific neuromodulators can also act directly at the level of the expression of plasticity to control its magnitude and polarity[34,35]. Aligned with this latter idea, disruptions of the serotonergic system or the noradrenergic system during ODP, respectively, prevent the closed-eye depression of the non-deprived eye response potentiation[36]. Our current findings point to an additional third level of control: retroactive modifications in a reinforcement-like manner.

The cortical modifications induced by monocular deprivation have been largely considered a form of unsupervised Hebbian learning in which cortical circuitry slowly becomes tuned to the statistic of the environment via the slow accumulation of small incremental Hebbian changes[37–39]. In this view, Hebbian rules are sufficient, and neuromodulation serves primarily to increase the gain of plasticity in a behavioral state-dependent manner. In contrast, the induction and retroactive transformation of eTraces suggest a more punctuated pace of changes, with neuromodulators restricting the plastic changes only to moments of valuable biological experience. Although our results do not rule out an independent contribution of unsupervised-like plasticity, it must be noted that reinforced learning can be highly "efficient"[2]. For example, 60 NE pairings within one hour induce a shift of

magnitude comparable to that obtained after 7 days of monocular deprivation[22]. In contrast to unsupervised learning, during reinforced learning only a small proportion of pre- and postsynaptic coincidences evoked by vision result in synaptic plasticity. Considering that synaptic plasticity could be a metabolically demanding phenomenon[40–42], restricting and subordinating visual cortical plasticity to behavioral relevance could be advantageous (i.e. Li et al.)[43]. Finally, we would like to note that the demonstration of reinforcement-like learning via the transformation of eTraces in V1, somewhat unexpected in a primary sensory cortex, suggests the universality of the rules governing cortical plasticity. Indeed, the primary visual cortex is increasingly considered not merely an early stage of visual processing but also a primary locus of perceptual learning driven by behavioral relevance variables including reward, fear, attention, saliency, novelty[44–51] in which neuromodulatory systems likely play a central roles[2,52,53].

Most theoretical work and most of the experimental demonstrations of eTraces have focused on the case for LTP only. Complementary to the potentiation studies and consistent with a prior in vitro study[15] demonstrating distinct traces for LTP and LTD, our results revealed that visual cortical activity generates eTraces that can be converted into LTP and LTD by NE and 5HT. Importantly, responses to the same stimulus were potentiated or depressed depending on whether it was paired with NE or 5HT. This indicates that visual activity generates both traces at the same time, and the results of single-cell experiments (Fig. 3) indicate coexistence in the same cells. Whether both eTraces can coexist in the same synapses remains an intriguing question. Another unanswered question is the identity of the molecular mechanism underlying the induction and conversion of the eTraces. A plausible suggested scenario is that the LTP and LTD eTraces represent the residual activity of kinases and phosphatases, respectively, and that β-AR and 5HT$_{2C}$R modify the phosphorylated state of the AMPAR promoting/enabling their trafficking in and out of the synapse[15]. Independent of the molecular mechanisms, the existence of distinct eTraces for LTD in addition to those for LTP has important consequences. Theoretical studies showed the presence of the two distinct eligibility traces enables learning reward timing and magnitude via the competition of the two traces[15,54]. It is also worth noting that the opposite/complementary function of 5HT and NE in visual cortical plasticity reported here, parallels the duality of actions proposed for 5HT and dopamine Daw et al.[55] and resonates with proposed roles for serotonin in cognitive flexibility[56,57], supporting the generality of the two eTraces system as a mechanism of plasticity.

## Methods
**Animals**. All protocols were approved by the Institutional Animal Care and Use Committee (IACUC) at Johns Hopkins University and followed the guidelines established by the Animal Care Act and National Institutes of Health (NIH). In Figs. 1–4 and Supplementary Figs. 1–4, the mice at the age of P40–P80 were used. NE-ChR2 mice were produced by crossing THi-cre homozygote (provided by Dr. Jeremy Nathan) with Floxed-ChR2 (B6; 129S-Gt(ROSA)26Sortm32(CAG-COP4*H134R/EYFP)Hze/J (Jackson Laboratory, Bar Harbor, ME). For the 5HT-ChR2 mice, Tph2-ChR2 (B6;SJL-Tg(Tph2-COP4*H134R/EYFP)5Gfng/J) (Figs. 1, 2 and 4, Supplementary Figs. 1, 2 and 4) (Jackson Laboratory) or Sert-Cre mice (B6.129(Cg)-Slc6a4tm1(cre)Xz/J) (Fig. 3 and Supplementary Fig. 3) (Jackson Laboratory) were used. In Figs. 5, 6 and Supplementary Fig. 5, C57BL/6J (Jackson Laboratory) mice at the age of P28–P30 (juvenile) or of P90–P96 (young adult) were used. Mice were reared in a 12 h light/dark cycle.

## Slice electrophysiology
*Preparation of cortical slices*. Brain slices from mice (4–5 weeks) were prepared as described previously[35]. Briefly, mice were anesthetized using isoflurane vapors, then immediately decapitated. The brain was removed and immersed in the ice-cold dissection buffer (dissection buffer in mM: 212.7 sucrose, 5 KCl, 1.25 NaH$_2$PO$_4$, 10 MgCl$_2$, 0.5 CaCl$_2$, 26 NaHCO$_3$, and 10 dextrose bubbled with 95% O2/5% CO$_2$ (pH 7.4)). Thin (300 µm) coronal slices of visual cortex or piriform cortex were cut in the ice-cold dissection buffer and transferred to a light-tight holding chamber with artificial cerebrospinal fluid (ACSF in mM: 119 NaCl, 5 KCl, 1.25 NaH$_2$PO$_4$, 1 MgCl$_2$, 2 CaCl$_2$, 26 NaHCO$_3$, and 10 dextrose bubbled with 95% O2/5% CO$_2$ (pH 7.4)). The slices were incubated at 30 °C for 30 min and then kept at room temperature until they were transferred to the disrupting peptide incubation chamber or to the recording chamber. For the disrupting peptide pre-incubation, the slices were incubated in the ACSF containing disrupting peptides (10 µM) at least for 15 min.

*Whole-cell current-clamp recordings*. Recordings were made from layer 2/3 pyramidal neurons in normal ACSF with glass pipettes (3–5 Mohm) filled with potassium-based internal solution (internal solution in mM: 130 K-gluconate, 10 KCl, 0.2 EGTA, 10 HEPES, 4 Mg-ATP, 0.5 Na-GTP, and 10 Na-phosphocreatine (pH 7.2–7.3, 280–290 mOsm)). 500 ms square wave current step was injected to evoke 5-6 spikes every 10 s and the number of spikes was monitored throughout the recording. After 2-3 min of baseline recording, Salbutamol (40 µM) or Ro 60-0175 (10 µM) was perfused.

## Cranial windows and other surgery
*Surgery for optical imaging*. Intrinsic signal imaging was performed through a glass cranial window prepared as described previously[58] with some modifications. Briefly, 4–6 weeks old mice were anesthetized with isoflurane (2–3% for induction; 1.5% for maintenance in oxygen) and placed at a stereotaxic frame. Dexamethasone (4.8 mg/kg, i.m.) and atropine (0.05 mg/kg, s.c.) were administrated to prevent brain edema and mucosal secretion, respectively. The skull was exposed and washed with hydrogen peroxide. The center coordinate of V1 was marked [−3.6/2.5] (A/P, M/L) and a 3 mm diameter craniotomy was performed using a dental drill. Following the removal of the bone flap, a three-layered glass window[58] was inserted and fixed with dental cement (C&B metabond, Parkell Inc., NY). All procedures were performed under red LED to avoid the risk of activation of the terminal ChR2. After the surgery, the glass cranial window was covered with non-transparent silicone sealant (Kwik cast; World Precision Instruments, Sarasota, FL). The mice received an injection of Meloxicam (5 mg/kg, s.c.) and Carprofen (70 µg/ml) in the drinking water for 7 days following the surgery was given as an analgesic. The mice were used for the experiment after at least 10 days of recovery period.

*Surgery for in vivo whole-cell recordings*. For the preparation of cranial window on top of the V1, the mice were anesthetized with urethane (i.p., 1.2% in saline) and supplemented with isoflurane (0.5–1.2% in oxygen). Atropine was injected subcutaneously to reduce mucosal secretion (0.05 mg/kg) and eye drops were administered to keep eyes moist. The skull was exposed and washed with 3% hydrogen peroxide. A head bar was attached to the anterior region of the head using dental cement (C&B metabond). The location of V1 was identified by stereotaxic coordinate (A/P: −3.6, M/L: 2.5). For initial trials, the location was confirmed using ISI. A small (~0.5 mm) cranial window was made with a dental drill under red LED light and covered with 1% low melting point agarose (A9793, Sigma-Aldrich, St. Louis, MO) in the modified ASCF (in mM: 140 NaCl, 2.5 KCl, 11 Glucose, 20 HEPES, 2.5 CaCl$_2$, 3 MgSO$_4$, 1 NaH$_2$PO$_4$). Dura was not removed.

*Surgery for two-photon imaging*. The glass cranial window and the metal headpost to fixate the head were installed as described in Goldey et al.[58]. To express GCamp6f, 50 nl of AAV9-CamKII-Gcamp6f-WPRE-sv40 (Addgene) was injected in 2–3 places near the central coordinate of the V1 [−3.6/2.5] (A/P, M/L), at a depth of 50 um.

*Preparation of the 5HT-ChR2 mice for in vivo whole-cell and two-photon recordings*. We found that in the whole-cell experiments, the direct illumination of V1 to release 5HT resulted in a low success rate of response depression. Hence, we switched to the more effective method of direct illumination of the Raphe nuclei as done in the Cohen lab, which uses the Sert-Cre line[59]. Therefore, after successful pilot experiments, we adopted this approach and line for subsequent experiments. In these experiments, 5HT-ChR2 mice were produced by viral expression of Cre-dependent ChR2 in the dorsal raphe nucleus (DRN) of Sert-Cre mice. Injection of AAVs (rAAV5-EF1a-DIO-hChR2(H134R)-EYFP) (Addgene, Watertown, MA) was performed with an angled approach (16°) in three different coordinates: [−4.66/−2.8], [−4.6/−3], [−4.54/−3.32] (A/P, D/V). For the installation of the optic fiber (200 µm) (Thorlabs, Newton, NJ), a craniotomy was made at the coordinate of [−4.6/1.33] (A/P, M/L). The optic fiber was placed with a 20° lateral angle targeting at the coordinate of [−4.6/0.2/−3.1] (A/P, M/L, D/V). Mice were used for the recording experiment at least 2 weeks after the viral injection.

*Intracerebroventricular infusion of disrupting peptides*. Cell membrane permeable peptide DSPL (Myr-QGRNSNTNDSPL) and its control peptide DAPA (Myr-QGRNSNTNDAPA) (gifts from J.W.H.) were prepared in 10% Dimethyl sulfoxide (DMSO)/90% ACSF, whereas the 2C-Ct (TAT-VNPSSVVSERISSV) peptide and its control peptide CSSA (TAT-VNPSSVVSERISSA) (purchased from GenScript (Piscataway, NJ)) were prepared in ACSF. Each disrupting peptide was injected via cannula using a syringe pump (2 ul, 660 uM) or using a subcutaneously installed osmotic minipump (Alzet 1007D; Durect Corp., Cupertino, CA) combined with Brain Infusion Kit (Durect Corp.) (12 ul/day, 150 uM).

*Monocular deprivation*. After the mice were anesthetized with isoflurane (2-3% for induction; 1.5% for maintenance), the margins of upper and lower eyelids were trimmed and sutured shut. A small amount of Neosporin was applied to the sutured eye to prevent an infection. The mouse was given an injection of Meloxicam (5 mg/kg, s.c.) after the surgery. The sutured eye was checked before it was opened for an imaging session to make sure the integrity of the lid suture.

**Visual stimulation and visual conditioning**. All visual stimulation was presented on an LCD monitor screen diagonally placed 25 cm from the right (contralateral) eye of mouse, except for the ocular dominance experiments where the screen was placed in front of the mouse.

*Intrinsic signal imaging of Vertical/Horizontal responses*. Visual stimulation consisted of a periodic vertical or horizontal drifting bar (2°) moving unidirectionally at 1 cycle/6 s temporal frequency. Each visual stimulation was presented for 5 min. The order of the visual stimuli presentation was arranged such that the first and fourth stimuli pair and second and third stimuli pair have the same orientations (i.e. direction order: (1) 270°, (2) 180°, (3) 0°, (4) 90°).

Visual conditioning consisted of 30 repetitions of visual stimulation blocks, in which square wave drifting gratings of 4 different directions (270°, 180°, 90°, 0°) with 1 cycle/20° spatial frequency and 1 cycle/6 s temporal frequency are presented for 5 s followed by 25 s blank screen sequentially. The train of photoactivation (10 ms pulse at 20 Hz for 5 s) followed the offset of the visual stimulus of the neuromodulator coupled orientation.

*Intrinsic signal imaging of ocular dominance*. Visual stimulation for the recording of the ocular dominance at the binocular region was restricted to the binocular visual field (−5° to +15° azimuth) and consisted of a horizontal thin bar (2°), continuously drifting for 5 min in upward (90°) and downward (270°) directions to each eye separately[35]. The sequence of the visual stimulation was arranged such that the first and fourth stimuli were for the same eye (270° and then 90°), and second and third for the other eye (90° and then 270°).

Visual conditioning consisted of 60 repetitions of visual stimulation blocks, in which square wave drifting gratings of 2 different directions (270°, 90°) were presented to the conditioned eye (right eye, contralateral to the recorded hemisphere) for 5 s followed by 55 s of a blank screen. The train of photoactivation (10 ms pulse at 20 Hz for 5 s) followed the offset of the visual stimulus of the neuromodulator coupled orientation.

*Visual stimulation and conditioning for whole-cell recordings*. Visual stimulation consisted of two non-overlapping rectangular flashing lights (500 ms) alternately presented every 5 s. To decide the location of the stimuli, the screen was divided into 15 subregions and two of the subregions that evoked reliable VEPSPs were selected at each experiment. The contrast of both stimuli was adjusted to evoke comparable subthreshold VEPSP and they were randomly assigned to serve as conditioned or non-conditioned stimuli. (In NE experiments: VEPSP conditioned = 3.65 ± 0.31 mV*sec; VEPSP non-conditioned = 3.18 ± 0.31; paired *t* test: *p* = 0.651, *n* = 19. In 5HT experiments: VEPSP conditioned = 3.85 ± 0.59; VEPSP non-conditioned = 3.76 ± 0.52; paired *t* test: *p* = 0.832, *n* = 12).

Visual conditioning consisted of 30 repetitions during which the two flashing lights at the selected subregion were alternated every 10 s and paired with square suprathreshold current pulses (400 ms) injected 100 ms before the onset of each visual stimuli such that the evoked spikes and VEPSPs overlap significantly. The amplitude of the current pulse determined to evoke at least 10 spikes. In addition, one of the two visual stimuli (chosen randomly: Neuromodulator Coupled, NC) was further paired with a 1 s train of blue light pulses (10 ms at 20 Hz) delivered immediately after the offset of the NC visual stimulation. The other visual stimulus (Neuromodulator Uncoupled, NU) was not further paired and served as an internal control.

*Two-photon stimulation and conditioning*. Baseline visual stimulation consisted of 16 repetitions blocks, each block consisting of square wave drifting gratings at 12 different orientations (1 cycle/20° spatial frequency and 3 Hz temporal frequency) and 2 blank stimuli. Each stimulus was presented in a pseudorandom order for 3 s followed by 5 s of a blank screen.

For the visual conditioning, the neuromodulator coupled (NC) orientation was selected at each mouse based on the distribution of orientation preference of the neurons recorded in the 'Before session'. The orientation chosen elicited a significant but submaximal visual response. The visual conditioning consisted of 30 repetitions of visual stimulation blocks, in which square wave drifting gratings of the NC orientation or the orthogonal orientation in two opposite directions are presented for 5 s followed by 25 s interstimulus intervals sequentially. The train of photoactivation (10 ms pulse at 20 Hz for 5 s) followed the offset of the visual stimulus of the neuromodulator coupled orientation.

**Recording and analysis**
*Acquisition and analysis of the intrinsic signal*. ISI was performed as described previously[26,60] with some modifications. Briefly, visual responses were acquired using a Dalsa 1M30 CCD camera (Dalsa, Waterloo, Canada). The surface

vasculature and intrinsic signals were visualized with LED illumination (555-nm and 610-nm, respectively). The camera was focused 600 mm below the surface of the skull. An additional red filter was interposed to the CCD camera and intrinsic signal images were acquired. The response amplitude of the intrinsic signal of an orientation and the HV ratio was calculated as follows: (1) The cortical response at the stimulus frequency was extracted by Fourier analysis and the two maps generated with the opposite direction of drifting bar were smoothed by 5 × 5 low-pass Gaussian filter and averaged to generate the intensity map; (2) a combined intensity map was generated by the sum of intensity map of the vertical and the horizontal orientation; (3) the region of interest (ROI) was defined by the region where the combined intensity is bigger than 40% of peak amplitude; (4) response amplitude of each orientation was computed by average of the intensity of all pixels in the ROI of each map; (5) HV ratio was calculated by the average of (H-V)/(H + V) of all pixels in the ROI, where H and V are the horizontal (H) and vertical (V) orientation, respectively. The response amplitude of the intrinsic signal from each eye and the ODI was also calculated by the same methods, but its ROI was defined at 30% of peak response amplitude of the smoothed intensity map from the ipsilateral eye, and the ODI was calculated by the average of (C-I)/(C + I) of all pixels in the ROI, where C and I are the contralateral (C) and ipsilateral (I) eye, respectively.

*Whole-cell current-clamp recordings*. Mouse body temperature was maintained at 37 °C with a heating pad and rectal probe and the heart rate was monitored throughout the experiment by electrocardiogram. A reference electrode was placed near the cranial window and submerged in the 1% agarose, which has been kept moist with the modified ACSF. Recordings were made with a Multiclamp 700B amplifier (Axon Instruments, Foster City, CA) using the blind patch-clamp technique[61]. Recording electrode (pipette resistance: 4–6 Mohm) with biocytin (1%) filled potassium-based internal solution (in mM: 130 K-gluconate, 10 KCl, 0.2 EGTA, 10 HEPES, 4 Mg-ATP, 0.5 Na-GTP, and 10 Na-phosphocreatine (pH 7.2–7.3, 280–290 m Osm)) was used. Electrodes were inserted into the brain perpendicular and advanced with a motorized micromanipulator (Sutter Instrument, Novato, CA) in 1 μm increment. The depth of the recorded cell was estimated based on the depth from the pia and only the cells between 100 and 450 μm below the pia were used for analysis. Electrophysiological recordings and visual stimuli were controlled using acquisition software packages Stage (http://stage-vss.github.io) and Symphony (http://symphony-das.github.io). After the acquisition of whole-cell configuration, membrane potential was initially set to −70 mV by injection of hyperpolarizing current (30–260 pA) and was not adjusted throughout the recording. Input resistance (Ri) was monitored with hyperpolarizing current steps (50 pA, 100 ms) throughout the recording. The bridge was balanced, and the liquid junction potential was not corrected. Sweeps were filtered at 2 kHz, sampled at 10 kHz, and analyzed with custom code running in Matlab (The Mathworks, Natick, MA).

Photoactivation to activate ChR2 was done via either the optic fiber placed in proximity of the recording cranial window (NE-ChR2) or the optic fiber implanted (5HT-ChR2). For the analysis of the VEPSPs, (1) voltage traces were smoothened by taking median over 50 ms window to eliminate the contamination by sporadic spikes during the visual response; and (2) the integrals of the voltage change during the flashing light illumination (500 ms) from the baseline (average voltage during 100 ms period prior to the onset of the flashing light) were calculated.

*Two-photon calcium imaging and data analysis*. Habituation of the mice to head fixation on a treadmill was done at least three times before the recording experiment. During the imaging session, the mouse freely moved on a treadmill and the locomotion was recorded using a quadrature encoder (US Digital, WA). Imaging was done on a custom-built two-photon microscope (Janelia MIMMS) using Chameleon Ultra II laser (Coherent Inc., CA) and an 8 kHz resonant scanner (Sutter Instruments, CA).

Images were acquired at ~30 fps using Scanimage 2018 (Vidrio)[62] and analyzed using custom scripts written in Matlab. After image alignment, the region of interests (ROIs) were manually selected to analyze the visual stimulus-responsive cells based on standard deviation and response to visual stimuli (Supplementary Fig. 4a). The response to visual stimuli was calculated at each pixel as below:

$$\frac{\text{mean}(F\_vistim) - \text{mean}(F\_blank)}{\text{std}(F\_blank)} \qquad (1)$$

where F_vistim and F_blank indicate the fluorescence intensity by visual stimuli and blank stimuli, respectively, across the acquisition period. A semi-automated algorithm determined the shape of the template that was then used to extract fluorescent traces. Pixels on the boundary of these ROIs served as a local neuropil estimate. Fluorescence over time was measured by averaging within the ROI, and the fluorescence traces were subtracted 0.7x neuropil estimate to correct for possible contamination by neuropil. The fluorescence traces during mouse walking (treadmill speed 0.75 cm/s) were not included in the analysis because locomotion itself generates non-visual neuronal activity. $\Delta F/F$ was calculated as $(F-F_0)/F_0$, where $F_0$ was the mean fluorescence intensity during a second prior to the onset of the drifting gratings. The relative response amplitudes to the drifting gratings at each orientation were determined as z-scores. The average of 16 consecutive traces

$\Delta F/F$ traces (locomotion excluded) were calculated as the mean or the median, for awake mice and anesthetized mice, respectively.

The optogenetic conditioning was performed one day after the initial measurements and the same ROI was located using the vasculature as a landmark and the alignment was done under visual guidance.

The orientation selectivity of the neurons was characterized as 1-CirVar, as recommended, where CirVar is the circular variance, which was calculated using the circular statistics toolbox in Matlab. A Kruskal–Wallis test ($p < 0.05$) was used to test if any of the orientation responses was significantly different from the rest of the orientations.

The preferred orientation of the orientation-selective neurons ($p < 0.05$) was determined as the orientation of the vector resulting from the summation of all vectors at each tested orientation.

**Immunohistochemistry.** The anesthetized mice were transcardially perfused with 10% neutral buffered formalin solution. Following perfusion, the brain was extracted and kept in the fixative solution overnight. The brain was sliced into 70 μm coronal sections and the slices were transferred to phosphate-buffered saline (PBS). Slices were then permeabilized (2% Triton X-100 in 0.1 M PBS) for 1 hr before incubation with 1 mg/ml streptavidin-488 (in 0.1 M PBS containing 1% Triton X-100) overnight at 4 °C. The slices were washed in PBS for an hour and mounted on a slide glass. Slides were coverslipped with the Prolong Gold anti-fade mounting solution with DAPI (Cell Signaling Technology, Inc., Danvers, MA) incorporated. Confocal images were taken on a Zeiss laser stimulated microscope 700.

**Statistical analysis.** Normality was determined by the Shapiro-Wilk test using Prism (GraphPad Software, San Diego, CA). Wilcoxon signed-rank test (Figs. 1–3 and Supplementary Figs. 1, 3, 5), Wilcoxon rank-sum test (Supplementary Figs. 2, 5), or paired *t* test (Fig. 4) were performed using MATLAB. Two-way ANOVA followed by post hoc Sidak's multiple comparisons test (Fig. 5) and One-way ANOVA (Figs. 3, 5) followed by post hoc Holm–Sidak test were performed using Prism. Data are presented as averages ± s.e.m. otherwise mentioned.

**Reporting summary.** Further information on research design is available in the Nature Research Reporting Summary linked to this article.

## Data availability

Source data are provided with this paper. Raw data are available from the corresponding author on reasonable request.

## Code availability

Custom codes are available from the corresponding author on reasonable request.

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

## Acknowledgements
The authors thank H. Shouval for his valuable comments in the writing. Supported by grants R01DA042038 to J.Y.C., R01 MH097887 and R01 AG055357 to J.W.H., R01-EY014882 to H.K.L., R01EY12124 and R01EY025922 to A.K.

## Author contributions
S.H., L.M., D.S., and C.D.G. collected, analyzed, and interpreted the imaging data, B.L. and J.W.H. made the DSPL and DAPA peptides, C.D.G. and J.Y.C. developed optogenetics in the 5HT-ChR2 mice, S.H., H.K.L., and A.K. wrote the manuscript.

## Competing interests
The authors declare no competing interests
