## [Peer Review File · Nature Communications]

Norepinephrine Potentiates and Serotonin Depresses Visual Cortical Responses by Transforming Eligibility TracesREVIEWER COMMENTS

Reviewer #1 (Remarks to the Author):

In this interesting paper, the authors investigate a possible physiological underpinning of eligibility traces for plasticity processes during reinforcement learning. This builds on a powerful series of studies, in part performed by the authors' own laboratory, in which they have shown that serotonin (5HT) and norepinephrine (NE) have the capacity to modulate LTP and LTD processes. Here, they take this research up one level, and demonstrate *in vivo* using optogenetics, intrinsic imaging and electrophysiology that NE and 5HT can potentiate or depress visual cortical responses, respectively. I appreciate the various angles by which they have approached the topic (e.g. spike-timing; orientation-tuning; OD plasticity) as well as the pharmacology to block the effects. I see no major issues that would preclude publication and I have no major concerns, except from a few minor comments or suggestions.

1. In Figure 3 the authors potentiate or depress visually-induced EPSPs by coupling them with post-synaptic burst firing and optogenetic neuromodulator release. This is a very nice experiment but I am a bit confused by the design. The two alternating visual responses do not necessarily result in EPSPs of the same amplitude. How did the authors control for the possibility that some responses may by themselves be more or less saturated and therefore exhibit a different potential for LTP or LTD? E.g. in Fig 3b, one could imagine that the response in the left square has less potential to be potentiated than the right. And could this phenomenon be the reason for the lack of depression in the U-VEPSPs in the 5HT-ChR2 mice? This also begs the question as to whether burst-pairing with visual stimuli always leads to depression or no change, or whether it could result in potentiation as well – alike Pawlak et al. *eLife* 2013; or Gambino and Holtmaat *Neuron* 2012.

2. For Fig. 4, have the authors tried to potentiate responses (i.e. orientations) that did not drive spikes? One would expect this to be possible, as in principle all orientations are represented on V1 dendrites (e.g. see El-Boustani et al. *Science* 2018). It would be interesting to see if subthreshold responses themselves could be turned into suprathreshold responses by neuromodulator conditioning.

3. There are a few experimental design strategies that remain somewhat puzzling. For Fig. 3 the authors chose a different 5HT-Cre transgenic line. What was the reason for this choice? Could this relate to the differential potential of 5HT to depress traces in the various experiments? For Fig. 4 the serotonin experiments was done in anesthetized mice. The authors only mention that they tried the experiment in awake and that this did not work. In addition, the effects of 5HT conditioning remain really small, and one could wonder if the conclusions related to orientation-tuning depression are meaningful. Could the authors more elaborately comment on these issues?

4. The results in Fig. 4i-j and o-p are not very clearly described. Perhaps the authors could explain this better in the result section.

5. There are quite a few typos in the manuscript.

Reviewer #2 (Remarks to the Author):

This is a well-designed and executed study, using multiple approaches to test the hypothesis that eligibility traces set up by visual stimuli, which the authors have previously demonstrated *in vitro*, are able to condition plastic changes to visual responses in cortical visual areas in the predicted direction when followed by an optogenetically driven reinforcement signal. In anesthetized mice, using optogenetic activation of noradrenergic (NE) pathways as the reinforcer they show that optogenetic activation delivered immediately after a horizontal drifting grating potentiates the visual response to

that grating recorded using optical imaging, but not the response to a non-reinforced vertical grating. Alternatively, optogenetic reinforcement of the same visual stimuli activating 5HT pathways depresses the response. In the extended figures (which are very helpful and I would prefer were just bundled in the original figures for ease of navigating, although I understand this is a journal style) they show that this is not stimulus specific but specific to visual circuitry connected to the contralateral eye. They then confirm the cellular mechanisms of the potentiation and depression of optical responses by uncoupling noradrenergic or serotonergic receptor anchoring to PSD95, respectively.

Using in vivo whole-cell patch clamp recordings of V1 neurons under anaesthesia, they demonstrate the Hebbian nature of the plasticity at the single cell level by pairing visual stimulation from two different locations with intracellular current injection, with one response potentiated or depressed by the specific neuromodulator optogenetically released in that animal, and the other non-reinforced. I found this remarkable - please make clearer in the text that these are responses from the same neuron in the same animal, assuming they were. Were the visual stimuli usually adjacent to each other on the screen (presumably because of the limited range of the cell receptive field)? Did you test for degree of independence of these responses by summation of both simultaneously?

These observations were then studied in head-fixed awake animals using two-photon calcium imaging. They found they were able to alter the preferred orientation of cells imaged within the optical field using NE reinforcement but interestingly were unable using this preparation to shift the response away from the preferred following 5HT reinforcement – this required anaesthetised animals for some unexplained reason (please clarify the change in preparation within the figure legend for Fig 4 j-p).

Although the finding that optogenetic activation of circuits that are associated with visual reinforcement produce the changes that they predicted from their in vitro work, they miss the opportunity to demonstrate that in their hands the optogenetic activation they use is behaviorally reinforcing in the whole awake animal. Demonstration that the neuromodulatory stimulus supported reinforcement of behavior was taken in the only other previous in vivo STDP reinforcement plasticity study the authors cite, and this study should do the same. Alternatively, there is no demonstration in this paper that natural reinforcers will produce the effects that the authors demonstrate on visual responsiveness. In the discussion, they reference studies that have used rewards to demonstrate reinforcement of visual responses. At least one or other of these approaches should be included to fully convince the reader that the observations they report play a role in visual plasticity in more physiological situations of natural reinforcement.

Finally, they elegantly show that the postsynaptic coupling of NE and 5HT neuromodulator mechanisms are necessary for ocular dominance plasticity following monocular deprivation to depress the responsiveness in the circuitry connected to the deprived eye by 5HT mechanisms and to facilitate delayed potentiation in the circuitry connected to the non-deprived eye by NE mechanisms. This is important information validating the plasticity mechanisms they have elucidated in a more natural physiological modulation paradigm. However, I find it a stretch to marry these experiments with the preceding work to which the title of the paper relates, which is related to eligibility traces triggered by pre and post synaptic activity “which decay over seconds” being transformed into synaptic weight changes by a neuromodulator presented within this time window, following the trace induction.

Overall, the paper is well written and accessible, the figures are excellent and the results convincing. The text does need proofing throughout by a native English speaker to fix typos, ensure tenses are correct, and to ensure verbs match subjects in number; this was particularly an issue with the Methods section.

Reviewer #3 (Remarks to the Author):

Previously, the authors demonstrated the in vitro existence of eligibility traces (eTraces) in V1 slices. In this paper, they further investigated in vivo evidence of the eTraces in several scenarios of visual plasticity. There are extensive results being described here, using a variety of techniques, intrinsic imaging, optogenetics, peptide blocker infusion, in vitro recording, and 2-photon imaging etc. However, the methodology and data analysis should be better designed/described to support the conclusion.

The major concerns.

1. The optogenetic release of 5HT is not on V1. Given 5HT is widely distributed across brain, the activation might have multiple effects on different brain regions that confound the experimental results and interpretation.
2. Visual conditioning was properly done in only NE-ChR2 mice but not 5HT-ChR2 mice (Fig1-2, ext Fig1). This is particularly important given they failed to show the 5HT depression in awake animals (Fig 4). Authors should examine if in 5HT-ChR2 mice, pairing vertical bar with optogenetic activation of 5HT-ChR2 can still cause depression of overall response and whether the effect is restricted to the contralateral eye only (similar to ext Fig1).
3. Similarly, the drug infusion of 2C-ct should have a control of scrambled version of the peptide instead of non-injection. This is mentioned in the Methods and later section.

The minor concerns.

1. The in vivo effect of peptide infusion is not well addressed. Instead of in vitro experiment in ext Fig 2, LFP can be recorded in awake animals.
2. Methods section is very messy- it's converted from a word doc with comments and things that some analyses are missing. For example, 2-photon imaging section doesn't explain how each neuron is being matched before and after conditioning and circular variance is not described.
3. Figure 4i is not being discussed in the result or methods section.

We are grateful for having the opportunity to revise our study “Norepinephrine Potentiates and Serotonin Depresses Visual Cortical Responses by Transforming Eligibility Traces”. We are also grateful to the reviewers for the positive reception and for their valuable comments. We thoroughly revised the manuscript and made substantial changes to answer the issues addressed by the reviewers. Some of changes include new analysis and experiments. In particular, in response to Reviewer #1’s comments we did the requested experiments in the 5HT-ChR2 mice, which we agree were necessary to complete the picture. Reviewer#1 and #2 mostly raised issues of clarification and interpretation that we have addressed in the text. Reviewer#2 indicated the necessity of a behavioral demonstration of eTraces playing a role in reward-driven learning. We clarify now that this was not the central aim of the study. Rather it is the novel demonstration that ocular dominance plasticity is not a “passive” Hebbian processes, as largely assumed, but a reinforcement-driven process. The exact contribution of the many reinforcement signals (reward, novelty, fear, saliency, attention) remain to be determined.

We hope that with these improvements the study will be acceptable for publication in Nature Communications. The answers to the specific comments of the reviewers are listed below in bold; changes in the manuscript are highlighted in red.

Reviewer #1:

In this interesting paper, the authors investigate a possible physiological underpinning of eligibility traces for plasticity processes during reinforcement learning. This builds on a powerful series of studies, in part performed by the authors’ own laboratory, in which they have shown that serotonin (5HT) and norepinephrine (NE) have the capacity to modulate LTP and LTD processes. Here, they take this research up one level, and demonstrate in vivo using optogenetics, intrinsic imaging and electrophysiology that NE and 5HT can potentiate or depress visual cortical responses, respectively. I appreciate the various angles by which they have approached the topic (e.g. spike-timing; orientation-tuning; OD plasticity) as well as the pharmacology to block the effects. I see no major issues that would preclude publication and I have no major concerns, except from a few minor comments or suggestions.

1. In Figure 3 the authors potentiate or depress visually-induced EPSPs by coupling them with post-synaptic burst firing and optogenetic neuromodulator release. This is a very nice experiment but I am a bit confused by the design. The two alternating visual responses do not necessarily result in EPSPs of the same amplitude. How did the authors control for the possibility that some responses may by themselves be more or less saturated and therefore exhibit a different potential for LTP or LTD? E.g. in Fig 3b, one could imagine that the response in the left square has less potential to be potentiated than the right.

We now state more clearly in the methods that the stimulus contrast was adjusted to obtain subthreshold VEPSPs of comparable magnitude in both stimuli. For example, in the 19 NE experiments, the average Conditioned VEPSP was 3.65 ± 0.31 mV*sec, whereas the average Non-conditioned VEPSP was 3.18 ± 0.31 (paired t-test: $p=0.651$). Similarly, in 5HT-ChR2 mice the initial magnitude of the VEPSPs was also comparable in both cases (conditioned = 3.85 ± 0.59 ; non-conditioned = 3.76 ± 0.52 ; paired t-test: $p=0.832$, $n=12$)

And could this phenomenon be the reason for the lack of depression in the U-VEPPs in the 5HT-ChR2 mice?

Besides being of comparable magnitude, conditioned and non-conditioned stimuli were assigned randomly

This also begs the question as to whether burst-pairing with visual stimuli always leads to depression or no change, or whether it could result in potentiation as well – alike Pawlak et al. eLife 2013; or Gambino and Holtmaat Neuron 2012.

Burst pairing uncoupled from neuromodulator release lead to no change in the 5HT-ChR2 mice (Fig3g) and a modest depression in NE-ChR2 mice (3i). This depression might represent heterosynaptic depression (akin to that described by WC Abraham in the hippocampus) or could result from activation of α 1 adrenoreceptors that exhibits more affinity and less desensitization than β -adrenoreceptors.

The burst overlapped with the synaptic response for hundreds of milliseconds. This design was chosen to ensure a “neutral STDP-like” paradigm with no net pre-post or post-pre situation. It worked, as revealed by the absence of changes in Fig. 3 i

2. For Fig. 4, have the authors tried to potentiate responses (i.e. orientations) that did not drive spikes? One would expect this to be possible, as in principle all orientations are represented on V1 dendrites (e.g. see El-Boustani et al. Science 2018). It would be interesting to see if subthreshold responses themselves could be turned into suprathreshold responses by neuromodulator conditioning.

In the NE-ChR2 mice, 7 out of 57 cells did not respond to the conditioned orientation (but responded to other orientations), 4 of these initially “silent” cells became responsive to the conditioned orientation. Since we measured changes in the soma this outcome is consistent with the idea posed by the reviewer. Our sample size is too small, however, to support that conclusion with statistical significance.

3. There are a few experimental design strategies that remain somewhat puzzling. For Fig. 3 the authors chose a different 5HT-Cre transgenic line. What was the reason for this choice?

In the methods section, we now explain why

“Preparation of the 5HT-ChR2 mice for in vivo whole-cell and two-photon recordings

We found that direct illumination of V1 to release 5HT yielded highly inconsistent results in the whole-cell experiments, therefore we switched to direct illumination of the Raphe nuclei as done in the Cohen lab, which uses the Sert-Cre line⁶¹.

Could this relate to the differential potential of 5HT to depress traces in the various experiments? For Fig. 4 the serotonin experiments was done in anesthetized mice. The authors only mention that they tried the experiment in awake and that this did not work.

It is unclear to us why it did not work in the awake mice. One plausible scenario is that the 5HT levels are high in the resigned head-fixed awake mice. We tried the anesthetized preparation because it worked fine for the depression of the responses to vertical and horizontal bars (fig1,2). The goal of these experiments was to test whether the timed pairing of neuromodulators shifts the orientation selectivity of the cells.

In the results section we now state:

“It is unclear whether this is due to elevated levels of 5HT or an elevated threshold for depressing mechanisms in the awake mice.”

In addition, the effects of 5HT conditioning remain really small, and one could wonder if the conclusions related to orientation-tuning depression are meaningful. Could the authors more elaborately comment on these issues?

Some measures of changes after 5HT (60% reduction in Fig. 4m) might look small compared to the dramatic increase after conditioning with NE (>300% Fig 4f), but they are highly significant and larger than other plastic changes reported in the cortex. For example, changes in synaptic quantal size after deprivation are typically no larger than 20% (see any Turrigiano paper, for instance). Also, the aim of the experiment was to test whether the mechanism can be recruited in vivo, not to maximize the outcome.

4. The results in Fig. 4i-j and o-p are not very clearly described. Perhaps the authors could explain this better in the result section.

The results now reads:

“In the subset of 31 clearly orientation-selective cells, the preferred orientation shifted towards the conditioned one (Fig. 4h), without losing overall selectivity (Fig. 4i), characterized as 1-CirVar (see methods)”

The legend for Fig 4 now reads

“h, Angular difference between the preferred orientations and the conditioned orientation measured before and after the visual conditioning diminished. This analysis includes only oriented cells with initial preferred direction significantly different from the conditioned one. i, 1-CirVar as a measure of orientation selectivity of the cells measured before and after visual conditioning (see methods)”

The methods now reads.

“The orientation selectivity of the neurons was characterized as 1-CirVar, as recommended, where CirVar is the circular variance, which was calculated using the circular

statistics toolbox in Matlab . A Kruskal-Wallis test ($p < 0.05$) was used to determine to test if any of the orientation responses is significantly different from the rest of the orientations”

5. There are quite a few typos in the manuscript.

the text was now checked by Grammarly and a native speaker

Reviewer #2 (Remarks to the Author):

This is a well-designed and executed study, using multiple approaches to test the hypothesis that eligibility traces set up by visual stimuli, which the authors have previously demonstrated in vitro, are able to condition plastic changes to visual responses in cortical visual areas in the predicted direction when followed by an optogenetically driven reinforcement signal. In anesthetised mice, using optogenetic activation of noradrenergic (NE) pathways as the reinforcer they show that optogenetic activation delivered immediately after a horizontal drifting grating potentiates the visual response to that grating recorded using optical imaging, but not the response to a non-reinforced vertical grating. Alternatively, optogenetic reinforcement of the same visual stimuli activating 5HT pathways depresses the response. In the extended figures (which are very helpful and I would prefer were just bundled in the original figures for ease of navigating, although I understand this is a journal style) they show that this is not stimulus specific but specific to visual circuitry connected to the contralateral eye. They then confirm the cellular mechanisms of the potentiation and depression of optical responses by uncoupling noradrenergic or serotonergic receptor anchoring to PSD95, respectively.

Using in vivo whole-cell patch clamp recordings of V1 neurons under anaesthesia, they demonstrate the Hebbian nature of the plasticity at the single cell level by pairing visual stimulation from two different locations with intracellular current injection, with one response potentiated or depressed by the specific neuromodulator optogenetically released in that animal, and the other non-reinforced. I found this remarkable - please make clearer in the text that these are responses from the same neuron in the same animal, assuming they were.

We clarify in the text that we recorded both conditioned and non-conditioned responses in the same cell.

Were the visual stimuli usually adjacent to each other on the screen (presumably because of the limited range of the cell receptive field).?

Yes, the visual stimuli were always adjacent. Our experimental paradigm was loosely based on a previous in vivo STDP paper that tested 4 (large) subfields, which often yielded only one responsive subfield (Neuron 2006. 49:183). To increase the chances of getting at least 2 equally responsive subfields we further subdivided into 15 subfields. Perhaps further subdivisions might allow non-adjacent fields, but the time required for presenting all the additional stimuli needed for this becomes prohibitively long for in vivo whole-cell recordings.

Did you test for degree of independence of these responses by summation of both simultaneously?

We did not test for independence. Although that is a standard test in slice experiments, it is rather difficult to add that for in-vivo whole-cell experiments, which are quite limited by the duration we can hold the cells for stable recording. Nevertheless, we note that only the conditioned stimuli potentiated in the NE experiments and depressed in the 5HT experiments, which allowed for a straightforward interpretation.

These observations were then studied in head-fixed awake animals using two-photon calcium imaging. They found they were able to alter the preferred orientation of cells imaged within the optical field using NE reinforcement but interestingly were unable using this preparation to shift the response away from the preferred following 5HT reinforcement – this required anesthetised animals for some unexplained reason (please clarify the change in preparation within the figure legend for Fig 4 j-p).

We now state:

“It is unclear whether this is due to elevated levels of 5HT or an elevated threshold for depressing mechanisms in the awake mice.”

Please, see the response to the reviewer#1 who posed the same question

Although the finding that optogenetic activation of circuits that are associated with visual reinforcement produce the changes that they predicted from their in vitro work, they miss the opportunity to demonstrate that in their hands the optogenetic activation they use is behaviorally reinforcing in the whole awake animal. Demonstration that the neuromodulatory stimulus supported reinforcement of behavior was taken in the only other previous in vivo STDP reinforcement plasticity study the authors cite, and this study should do the same. Alternatively, there is no demonstration in this paper that natural reinforcers will produce the effects that the authors demonstrate on visual responsiveness. In the discussion, they reference studies that have used rewards to demonstrate reinforcement of visual responses. At least one or other of these approaches should be included to fully convince the reader that the observations they report play a role in visual plasticity in more physiological situations of natural reinforcement.

We regret the confusion, the central aim of the study was the mechanisms of ocular dominance plasticity (ODP), not the mechanisms of reinforcement in reward-based cortical learning. Motivated by recent examples of reward-based reinforcement and perceptual learning in V1 we tested whether a retroactive action of neuromodulators -via conversion of eligibility traces- contributes to ODP. We show that these retroactive mechanisms are operational in vivo and that manipulations that specifically disrupt these mechanisms also disrupt ODP. We believe that this is an important point because ODP, the canonical model of cortical plasticity conducive to amblyopia, has been largely considered a “passive” form of learning. We have made changes in the introduction and discussion to clarify this point.

We eliminated a paragraph that emphasized the potential value of the disrupting peptides in the study of reward-driven visual learning. That probably contributed to the perception that the goal of the paper was to clarify perceptual learning.

Reinforcement-like learning is not restricted to reward, and perceptual learning can be also driven by fear, novelty, salience, and attention. It remains to be determined how each of these different “behavioral values” contribute to ODP. That is going to be a challenging task because the most defining aspect of ODP is the depression to the deprive-eye responses and most, if not all, models of reinforcement emphasize LTP-like changes, not LTD. We plan to begin addressing this issue using the recovery from long-term monocular deprivation as a model, which likely requires reinforced LTP of the deprived eye. But due to the scope of work needed, this will need to be a follow-up project.

Finally, they elegantly show that the postsynaptic coupling of NE and 5HT neuromodulator mechanisms are necessary for ocular dominance plasticity following monocular deprivation to depress the responsiveness in the circuitry connected to the deprived eye by 5HT mechanisms and to facilitate delayed potentiation in the circuitry connected to the non-deprived eye by NE mechanisms. This is important information validating the plasticity mechanisms they have elucidated in a more natural physiological modulation paradigm. However, I find it a stretch to marry these experiments with the preceding work to which the title of the paper relates, which is related to eligibility traces triggered by pre and post synaptic activity “which decay over seconds” being transformed into synaptic weight changes by a neuromodulator presented within this time window, following the trace induction.

True, we did not explore the parametrics of the timing of the reinforcement. Nevertheless, the experiments shown in fig 3 are consistent with a trace duration of less than 10 seconds; and those of figs 1 and 2, with duration shorter than 30 seconds.

Overall, the paper is well written and accessible, the figures are excellent and the results convincing. The text does need proofing throughout by a native English speaker to fix typos, ensure tenses are correct, and to ensure verbs match subjects in number; this was particularly an issue with the Methods section.

We made extensive re-arrangements corrections to the methods section. We hope it is clear now.

Reviewer #3 (Remarks to the Author):

Previously, the authors demonstrated the in vitro existence of eligibility traces (eTraces) in V1 slices. In this paper, they further investigated in vivo evidence of the eTraces in several scenarios of visual plasticity. There are extensive results being described here, using a variety of techniques, intrinsic imaging, optogenetics, peptide blocker infusion, in vitro recording, and 2-photon imaging etc. However, the methodology and data analysis should be better

designed/described to support the conclusion.

The major concerns.

1. The optogenetic release of 5HT is not on V1. Given 5HT is widely distributed across brain, the activation might have multiple effects on different brain regions that confound the experimental results and interpretation.

NE is also widely distributed, like dopamine and ACh. Both the Locus Coeruleus and the Raphe Nuclei are clear exemplars of diffusely projecting neuromodulatory systems. There are no 5HT or NE neurons projecting specifically to the visual cortex or any other cortex. In other words, optogenetic activation of NE or 5HT axons in V1 will not confine the release to V1 only. This lack of specificity is a well-known limitation of all neuromodulatory research.

The reason why the optogenetic release of 5HT and NE in figs 1 and 2 was done by direct illumination of V1 was experimental convenience. It was simpler that way. Since that approach did not work for the single-cell measurements (figures 3 and 4) we switched to Raphe stimulation.

2. Visual conditioning was properly done in only NE-ChR2 mice but not 5HT-ChR2 mice (Fig1-2, ext Fig1). This is particularly important given they failed to show the 5HT depression in awake animals (Fig 4). Authors should examine if in 5HT-ChR2 mice, pairing vertical bar with optogenetic activation of 5HT-ChR2 can still cause depression of overall response and whether the effect is restricted to the contralateral eye only (similar to ext Fig1).

We agree with the importance of these experiments with 5HT-ChR2 mice. We did these experiments and the results are now reported in extended figure 1 and 2.

3. Similarly, the drug infusion of 2C-ct should have a control of scrambled version of the peptide instead of non-injection. This is mentioned in the Methods and later section.

We also agree with the reviewer and the results with the control scrambled peptide are shown in figure 2 now (replacing those with no injection)

The minor concerns.

1. The in vivo effect of peptide infusion is not well addressed. Instead of in vitro experiment in ext Fig 2, LFP can be recorded in awake animals.

We moderated the conclusions and the sentence now reads

“Although we cannot exclude off-target effects of the peptides, the results are consistent with the notion that the visual stimulation-induced eTraces were subsequently converted into LTP and LTD by the retroactive action of NE and 5HT, respectively.”

We are confident that such off-target effects are minimal. The intrinsic imaging results in Figures 5 and 6 show normal responses in the controls cases treated with the peptides.

2. Methods section is very messy- it's converted from a word doc with comments and things that some analyses are missing. For example, the 2-photon imaging section doesn't explain how each neuron is being matched before and after conditioning and circular variance is not described.

We now explain these results. Please see our answer to question#4 of reviewer #1 who raised the same issue.

3. Figure 4i is not being discussed in the result or methods section.

We now discuss this as detailed in our response to point#4 of reviewer#1”

REVIEWER COMMENTS

Reviewer #1 (Remarks to the Author):

I am very happy with the response of the authors to my comments and the revised version of the manuscript. I have no further comments.

Reviewer #2 (Remarks to the Author):

The revised manuscript is greatly improved. I am satisfied that the authors have clarified my concerns and added additional experiments to satisfy the concerns of the other reviewers.

Reviewer #3 (Remarks to the Author):

Overall, there are substantial revisions to address previous issues. However, even with significant improvement of the 5-HT, it is concerning that different 5HT mouse lines are used. If 5HT-ChR2 mice have issues while Sert-Cre line is a more reliable line, all experiments should be done in 5HT-ChR2 mice, because the inconsistency in the whole-cell recording in 5HT-ChR2 might indicate there are likely some issues about the circuit in these mouse line.

For example, a collected effort by Steinmetz et 2017 eNeuron has reported that some transgenic lines have seizures. So if that's the case for this 5HT-ChR2 mice line, it would be worrisome to use them for any research studying the cortical E/I balance, plasticity, or the general circuit questions.

We are very grateful for the positive reception of this revised version. Reviewers#1 and #2 were very positive, and Reviewer #3 raised new concerns about using two different 5HT-ChR2 mice lines. We clarify that issue in the methods section. We hope that with these improvements the study will be acceptable for publication in Nature Communications. The answers to the specific comments of the reviewers are listed below in bold; changes in the manuscript are highlighted in red.

Reviewer #1 (Remarks to the Author):

I am very happy with the response of the authors to my comments and the revised version of the manuscript. I have no further comments.

We thank the reviewer

Reviewer #2 (Remarks to the Author):

The revised manuscript is greatly improved. I am satisfied that the authors have clarified my concerns and added additional experiments to satisfy the concerns of the other reviewers.

We thank the reviewer

Reviewer #3 (Remarks to the Author):

Overall, there are substantial revisions to address previous issues. However, even with significant improvement of the 5-HT, it is concerning that different 5HT mouse lines are used. If 5HT-ChR2 mice have issues while Sert-Cre line is a more reliable line, all experiments should be done in 5HT-ChR2 mice, because the inconsistency in the whole-cell recording in 5HT-ChR2 might indicate there are likely some issues about the circuit in these mouse line.

We regret that we were not clear that the initial low yield of positive results in the whole-cell experiments was due to the approach, NOT the line used. Based on conversations with colleagues with expertise in serotonergic processes in the cortex, we reasoned that the illumination of the 5HT axons in V1 was insufficient for producing substantial optogenetic release needed for the single-cell experiments (it was OK for detecting “bulk” changes in figure 1 and 2, however). Therefore, we switched to the more effective approach of direct illumination of the raphe nuclei as done in the Cohen lab. There they use the Sert-Cre line, and after a few successful experiments with these mice, we decide to keep using that line for the internal consistency of the whole-cell experiments. The methods section now reads:

We found that in the whole-cell experiments, the direct illumination of V1 to release 5HT resulted in a low success rate of response depression. Hence, we switched to the more effective method of direct illumination of the Raphe nuclei as done in the Cohen lab, which uses the Sert-Cre line⁶¹. Therefore, after successful pilot experiments, we adopted this approach and line for subsequent experiments.

In addition, we do not see the need for repeating the experiments of figures 1,2 in additional 5HT lines. We demonstrated optogenetically-induced changes that were input-specific and were prevented by disrupting the anchoring of 5HT2c receptors. The magnitude of the effect might vary among lines, yet, the central point is that timely opto-illumination of a validated 5HT-ChR2 mouse line does depress visual cortical responses in a reinforcement-like manner.

For example, a collected effort by Steinmetz et 2017 eNeuron has reported that some transgenic lines have seizures. So if that's the case for this 5HT-ChR2 mice line, it would be worrisome to use them for any research.

Neither we nor other investigators (Neuron 103: 686; Molecular Brain 6: 59) have observed/reported evidence of seizures in this line. Moreover, the Steinmetz et al. study focused on GCaMP6 lines only, and there is no basis to presume that catecholaminergic lines are affected in the same manner.